# Eradicating the Scourge of Peste Des Petits Ruminants from the World

**DOI:** 10.3390/v12030313

**Published:** 2020-03-15

**Authors:** Felix Njeumi, Dalan Bailey, Jean. Jacques Soula, Bouna Diop, Berhe G. Tekola

**Affiliations:** 1Food and Agriculture Organization of the United Nations (FAO), Viale delle Terme di Caracalla, 00153 Rome, Italy; bounaa.diop@gmail.com (B.D.); berhe.tekola@fao.org (B.G.T.); 2The Pirbright Institute, Ash Road, Surrey GU24 0NF, UK; dalan.bailey@pirbright.ac.uk; 3World Organization for Animal Health (OIE), 12, rue de Prony, 75017 Paris, France; jj.soula@oie.int

**Keywords:** Peste des Petits Ruminants, control, eradication, challenges

## Abstract

Peste des Petits Ruminants (PPR) is a highly contagious viral disease of both domestic (goats and sheep) and wild ruminants. Caused by a morbillivirus, that belongs to the family Paramyxoviridae. The disease is clinically and pathologically similar to rinderpest of cattle and human measles. PPR is one of the most economically devastating viral diseases of small ruminants. In April 2015, the Food and Agriculture Organization of the United Nations (FAO) and the World Organisation for Animal Health (OIE) launched the PPR Global Control and Eradication Strategy (PPR GCES) with the vision for global eradication by 2030. There is a strong and lasting international consensus to eradicate the disease in order to protect the livelihoods of the world’s poorest populations. As with any disease, eradication is feasible when, policy, scientific and technical challenges are addressed. Ten majors challenges are described in this paper namely: understanding small ruminant production, facilitating research to support eradication, refining laboratory testing, improving epidemiological understanding of the virus, defining infection of wildlife and other species, optimizing vaccine delivery and novel vaccines, developing better control of animal movement, heightening serological monitoring, understanding socio-economic impact, and garnering funding and political will.

## 1. Introduction

### 1.1. The Disease and Its Impacts

Peste des Petits Ruminants (PPR) is a highly contagious and devastative viral disease of both domestic (goats and sheep) and wild small ruminants. It is caused by a morbillivirus and is clinically and pathologically close to rinderpest and human measles. PPR infection was first described in Côte d’Ivoire in 1942 and now countries in Africa, Asia and the Middle East have reported PPR infections [1]. Among the 198 countries recognized by the United Nations, 57 have already have PPR-free status as per May 2019 according to OIE Standards, while 67 are infected and 74 have never reported PPR. Infected countries are home to some 68% percent of the world’s 2.5 billion small ruminant according to FAO statistic in 2018, with this disease having a direct impact on over 300 million families who rely on small ruminants. Between 2008 and 2018, PPR socio-economic studies were carried out in several countries. During the 2006–2008 PPR outbreak, in Turkana of Kenya, over a million animals perished and the total value, in terms of loss of production, it was estimated to be USD 2.4 million [personal communication]. In 2017, Mongolia reported its first ever PPR outbreak in domestic small ruminants as well as in endangered saiga populations. In this case, spill-over of virus from livestock at a multiple locations and time points, led to subsequent spread among wild ungulates. Worryingly, estimates of saiga abundance suggested a population decline of 80%, raising substantial concerns for the species’ survival [2]. The economic losses associated with the saiga deaths was estimated at USD 7.27 million [personal communication].

### 1.2. Need for Eradication, Validated by the International Community & Partner Organisations

Generally speaking, the factors [1,3] that favored the eradication of small pox and rinderpest also apply for PPR, namely: a reliable, safe and effective live attenuated vaccine which confers long-term immunity against all strains after a single inoculation; the availability of simple and effective diagnostic tests; a short infectious period (with no persistence or carrier state); transmission only by close contact; no evidence for a wildlife reservoir with epidemiological significance, and a short survival time for the virus in the environment (personal communication summarized in Table 1). Importantly, this is combined with stakeholder partnership at all levels and strong political will to support the eradication campaign. Furthermore, the success of the rinderpest eradication is still present in the minds of farmers and animal health personnel having a positive effect on their attitudes to any equivalent programme.

## 2. The FAO/OIE Joint PPR Global Control and Eradication Strategy (PPR-GCES)

### 2.1. The Strategy

The Food and Agriculture Organization of the United Nations (FAO) and the World Organisation for Animal Health (OIE), adopted at the 2015 international conference in Abidjan, Côte d’Ivoire, the PPR- Global Control and Eradication Strategy (GCES). The PPR- GCES, is built off of the strong scientific, economic and social case for PPR control and eradication combined with these Organizations’ success story of a related morbillivirus of cattle, rinderpest virus (RPV) eradication. This was also possible thanks to the strong partnerships with Africa Union Interafrican Bureau for Animal Resources (AU-IBAR), International Atomic Energy Atomic (IAEA), national governments, Non Governmental Organisations(Veterinaires sans frontiere Suisse in Sudan, Terra Nuova in Somalia among others), research institutes and universities. These partners are involved in GCES’s formulation and implementation. Rinderpest eradication showed the importance of risk-based vaccination as well as extensive and continued surveillance to ensure the continued absence of disease through well planned disease management and the preparation of proper contingency plans for re-emergence or re-introduction of the virus. These factors and their understanding have been built into the FAO/OIE PPR GCES. For rinderpest, it was also considered of critical importance that the programme was time-limited, which put pressure on countries to establish clearly that they were free of disease. In a broader context, more focus will now be placed on communication including large-scale media campaigns, social marketing, dissemination of printed materials, and synchronizing education and entertainment [1].

The PPR GCES promotes a stepwise approach to eradication based on four stages (Figure 1), which also provide an overview of how the programme will operate. These stages correspond to a combination of decreasing levels of epidemiological risk and increasing levels of prevention and control, and comprise a multi-stage, multi-country process involving assessment, control, eradication and maintenance of PPRV-free status. This ranges from stage 1 (where the epidemiological situation is being assessed), to stage 4 (when the country can provide evidence that there is no virus circulation either at a zonal or national level, and is ready to apply for OIE-official PPR-free status). Control activities, including vaccination, are implemented in stage 2 while stage 3 corresponds directly to PPR eradication. Of note, to enter stage 4 vaccination must be suspended in order to facilitate epidemiological monitoring of disease. Implementation requires the concerted delivery of preparedness plans, capacity building, improved stakeholder awareness and engagement; as well as the establishment of appropriate legal frameworks [1,4].

### 2.2. Initial Implementation of the PPR GCES 

Regardless of the stage in which a country initially places itself, it is imperative that sufficient capacity is secured in 5 key areas so that the country can move, with confidence, to the next stages of control and eradication. These five technical elements are: i) provision of an adequate PPR diagnostic system, ii) development of a PPR surveillance system, iii) implementation of a PPR prevention and control system, iv) establishment of a legal framework system and v) ensuring adequate stakeholders’ involvement in the campaign. The PPR Monitoring and Assessment Tool (PMAT) is used to support countries/territories in conducting self-assessments of their current stage [4].

As described above, from the 198 countries targeted for freedom, 57 are already PPR free, 67 are infected and 74 never reported PPR. From these 74 countries, 12 are at risk of PPR infection. Therefore, the targeted for the programme are these at risk and infected countries for a total of 79 to be supported for freedom although countries at risk should also be assisted for freedom on the historical basis. From the above 79 countries assessed since 2015 using PMAT , 30 are at stage 1, 38 in stage 2, 5 in stage 3 and 6 in stage 4 [personal communication].

### 2.3. International Coordination of the PPR-GCES

PPR GCES was subsequently turned into the PPR Global Eradication Programme (PPR GEP). A FAO/OIE Secretariat was established in March 2016 in FAO-Rome, the body responsible for the preparation and overall coordination of the PPR GEP. In October 2016, FAO and OIE launched the first five-years (2017–2021) of PPR GEP, developed through an inclusive and peer-reviewed drafting process. The main role of the Secretariat is to provide overall strategic coordinated direction, as well as to develop cost-effective control methodologies, tools, guidelines, training materials and networks. The Secretariat is supported by an Advisory Committee (AC) which provides strategic guidance and oversight on the execution of the programme while also playing an important advocacy role with policy makers, donors, national veterinary services and livestock owners. The Global Research and Expertise Network (PPR GREN) was launched in April 2018 during a meeting hosted by the Joint FAO/IAEA Division of Nuclear Techniques in Food and Agriculture in Vienna. PPR GREN was established as a forum for scientific and technical consultations to foster a science-based and innovative debate on PPR. The United Nations (UN) Rome Based Agencies Permanent Representatives “Friends of PPR GEP” was established in 2018 to: i) advocate for the importance of PPR GEP as a global challenge which can contribute to the UN sustainable development goals by 2030, ii) support the Secretariat in fundraising, and, iii) advocate during statutory meetings of FAO, the International Fund for Agricultural Development (IFAD) and World Food Programme (WFP). These four groups (Secretariat, AC, GREN and Friend of PPR GEP) formed the governance of the programme. Independently, national and regional programmes to control PPR are also being developed in alignment with the PPR GCES. In this context the FAO and OIE are playing a leading role in identifying and harmonizing potential global, regional, sub-regional and country partners (reference centers, research institutions, regional economic communities, civil society organisations, countries and farmers, amongst others); pinpointing the most appropriate roles for these partners; and helping to develop the mechanisms for their interaction and co-ordination. The other bodies likely to become involved include those with wide-ranging technical, political and/or economic interests in PPR eradication [4].

### 2.4. The Global Vaccination Strategy for PPR Eradication

Each country will start with an epidemiological assessment to identify the risk areas (stage 1). The PPR-GCES envisage that risk-based, targeted mass vaccination (stage 2) of assumed, high-risk populations will be progressively replaced by focused vaccination (stage 3) as more concrete epidemiological information becomes available (indeed some countries may be in position to follow a focused approach from the start). Vaccination campaigns will be accompanied by regular and constant surveillance for clinical disease, including routine passive reports, active disease searches, disease investigations and follow-ups. The collected and analysed data will be a critical indicator of the relative success of the campaign in reducing disease incidence and mortality. The success of the strategy shall depend on the implementation of vaccination campaigns, accounting for husbandry practices, mobility and the periodicity of small ruminants’ population renewal. A study conducted in Mauritania [5] showed that the PPR global strategy prevented the largest number of deaths (9.2 million vs. 6.2 for a random strategy) and provided one of the highest economic returns among all strategies (Benefit-Cost Ratio around 16 vs. 7 for the random strategy). According to its current cost, identification would be a viable investment that could reduce the number of vaccine doses required by 20–60%. Whilst the implementation of the identification system is crucial for PPR control, its success depends also on a coordinated approach at the regional level based on epizone or ecosystem approaches. The identification tool (ear-notched instead of painting horn or skin) should be harmonized, as was the case with rinderpest. Other factors to be considered in the vaccination campaign include logistical (fuel for vehicles, maintenance of the cold chain etc.), personnel (time and missions to vaccinate animals) and vaccine wastage (doses given to already vaccinated animals). The logistics and manpower could have a large impact on the vaccination campaign, accounting for, in some cases, up to 70% of the total campaign costs.

## 3. Key Challenges to the Eradication Campaign Programme?

Fortunately, there are no major technical or scientific barriers to eradicating PPR that exist as for other viruses, e.g., the absence of efficacious vaccines, multiple serotypes and repeated recombination, or an established carrier state in asymptomatic animals. Nevertheless, the control of PPR poses a series of challenges that need to be systematically addressed. These are detailed below:

### 3.1. Understanding Small Ruminant Production

Small ruminants are often found in marginalized extensive production systems and/or produced by poor people, women and/or pastoralists with limited access to services. For these people, small ruminants are often their most important asset. At the national level, small ruminant lobbies often have limited access to decision makers or resources partners, reducing the attention given to PPR (and small ruminant health in general). Another challenge facing small ruminant production is the short cycle nature of production or annual turnover rate of 30% to be considered regarding the vaccination programme. This implies a very different cost benefit picture for veterinary services and household units in relation to investments in vaccines and other animal health inputs. It is therefore important for any strategy to engage the small ruminant owners in improving their own systems and enhancing private sector service delivery channels. This is also where the FAO has a comparative advantage since it has extensive multidisciplinary experiences and a mandate to work within rural and smallholder settings, integrating production and animal health issues, along with social and economic issues either in livestock only or mixed farming systems.

### 3.2. Research to Facilitate Eradication

The research requirements to support eradication have been reviewed extensively elsewhere [6]. Moving forward the necessary research to be commissioned will be indicated by a gap analysis and must be both pragmatic and cost-effective. This may include research on: socio-economic analysis; epidemiological studies, e.g., on disease transmission chains; identification of small ruminant populations acting as reservoirs and modelling of this; identifying the genetic determinants of virulence; diagnostic and surveillance methodology; molecular epidemiology; filter paper sampling for serology; rapid field tests; and, optionally, novel vaccines (marked and recombinant, e.g., vaccines able to distinguish infected from naturally infected [DIVA] animals). Socio-economic studies and epidemiology may assist for advocacy. Other studies in epidemiology will help informing vaccination strategies. Few of these researches will assist for early diagnosis at the field or transport of specimens in the absence of cold chain facilities. The research for diagnostic tests for understanding their suitability to be used for PPR investigation in non-typical animal hosts, and if needed, development of new tests that are fit for purpose. Filter paper can be used as a good support for transportation of dried fluid to the laboratory for serology, molecular detection and genotyping. Research can also assist for differential diagnostics for post-vaccination monitoring/reassurance. The existing vaccines have already been developed in a thermostable forms and production has started (such as in Ethiopia). Other areas of research worth highlighting include an apparent variation in disease severity, which has been observed between different species of goats. Detailed research is required to define this susceptibility and to examine whether the variation extends to specific features such as the duration of viral shedding in infected animals, factors which will contribute to the dynamics of PPR transmission [7,8]. Another example would be developing a deeper understanding of how small ruminants are bred, raised, moved, traded and slaughtered in PPR endemic countries. This will enable a broader appreciation of the lifestyle of the livestock keepers who own and depend on these animals. This knowledge will certainly assist to design effective intervention strategies for vaccination campaigns, and ensure effective advice is given to local veterinary services about appropriate control measures.

### 3.3. Role of Laboratory Testing in the Eradication Campaign

An extensive battery of tests exists for diagnosing PPR and identifying the causative agent, PPR virus (PPRV) [9]. Basic tests include agar gel immunodiffusion assays while more sensitive and sophisticated assays such as immunocapture ELISA and quantitative real-time PCR (qRT-PCR) are also available. It is understood that diagnostic tools are inevitable components of the four-stages of the PPR global strategy, from assessment to the post-eradication phase. However, these tools may not be suitable for all stages of PPR control and eradication. For instance, diagnostics such as ELISA could be used for mass screening of clinical and serum samples, whereas immuno-chromatographic tests can be used at the field level (i.e., pen-side tests). Work to define minimal performance characteristics of these diagnostic assays and to establish bench marking procedures for diagnostic networks is needed. Standardization of tools should include tests for confirming outbreaks, tracking molecular epidemiology, supporting diagnostics for use in the field (pen-side test) and serological monitoring of vaccinated flocks. Assays with higher sensitivity, such as competitive ELISA, qRT-PCR and Loop Mediated Isothermal Amplification (LAMP) are clearly important for early diagnosis of PPR and also, theoretically, during the late stages of eradication or when sampling atypical hosts, e.g., wildlife with suspected infection [10]. During the latter stages of any control programme, suspected/doubtful outbreaks will also have to be reconfirmed using multiple laboratory tests [2,8,9,10]. Defining when and how to implement an appropriate range of diagnostic tests remains a multifactorial challenge, dependent on assay availability, training needs and deficits, various cost-benefit analyses and the correct validation of the assays involved. More detailed genetic analysis could also be used as a tool to identify linkages between foci of circulating virus, especially which populations are acting as incubators and sustaining infection in their regions. All laboratories should be linked through national, regional or global networks to allow information sharing and technology transfer.

### 3.4. Improving Epidemiological Understanding of PPRV 

Epidemiological research is required to better understand the PPR transmission dynamics, in particular its spread and infectivity and the different roles of wildlife and livestock species, production systems, ecosystems and viral lineages in this process. The overall goal is to identify critical control points, and optimal methods for intervention at these points, to support effective management of eradication. One example is a need to support the evaluation of R_0_ values, in relation to the various lineages of PPR and the various ecosystems it affects. Along these lines, focus on targeted short-term studies and appraisals complemented by evidence of a fully functional disease reporting system (in order to properly quantify the effectiveness of vaccination) or a prolonged period of field work in one or more infected countries may help to generate some of the basic longitudinal data required for computer simulations. The expert opinion and experience of professionals working in infected areas and with local knowledge can also be used to inform computer modelling. The results of these epidemiological studies will likely be the main driver for decisions about what levels of control might practically be achieved, as well as the feasibility or otherwise, of timely eradication according to the stepwise approach described above.

In stage 1 of the PPR GCES [1], the assessment aims to define the epidemiological situation through the following: i) early detection of the appearance of the disease, ii) demonstrating the absence of clinical disease or infection, and iii) determining and monitoring the prevalence, distribution and occurrence of the disease or infection. The surveillance components to be considered include: passive surveillance including zero reporting, active surveillance (including targeted or risk-based surveillance), sero-surveillance, syndromic surveillance, participatory disease searches (PDS), wildlife, and abattoir surveillance. A functional surveillance programme requires the following: (i) capacity building for field personnel; (ii) sensitization of livestock value chain actors with extension/communication materials; (iii) development of good animal health data management systems for information reporting; (iv) supporting the disease intelligence surveys. Through PDS, countries or zones may be classified as either: (i) low risk (currently PPR free); (ii) high risk or “hot spots” or (iii) infected (endemic) areas. From the methodological point of view there are different considerations when designing surveillance systems or analyzing data aimed at assessing either the baseline level of exposure to PPR or detecting differences between sub-populations assumed to have different levels of PPRV exposure. Countries identified through PDS as low can move directly to stage 4. Those considered as high risk or infected should move to stage 2. Surveillance options vary from country to country and within country depending on the GCES (stages 1). PDS and other surveillance components should prepare the ground for vaccination campaign in stages 2 and 3.

As discussed above, when surveillance has detected PPR foci, genetic approaches could identify the linkages among these foci. Examples of recent successes in this area include the global monitoring of PPRV epidemiology. Sequencing and phylogenetic tree analysis tools have allowed researchers to genetically define PPRV into four lineages; three historically centered in Africa and a single lineage, lineage IV, confined to Asia. However, data collected over the last decade has illustrated the wide incursion of lineage IV from the Middle East into North Africa, as well as the Sudan and some neighboring countries including the Central African Republic and Cameroon.

At a smaller scale, rinderpest eradication highlighted the importance of local stakeholder involvement, e.g., directly involving livestock keepers in the process of tracing disease, as being critically important in disease monitoring. To this end pen-side test kits are being developed that will allow sick animals to be tested for PPR in the field, allowing much more rapid disease identification and therefore more rapid responses to outbreaks. The requirement for this local, and international approach was highlighted by Taylor and others in Oman [11] who raised the possibility that PPR could be maintained within urban goat populations with periodic epidemic extension into rural areas as populations of susceptible young stock build up, akin to cases of human measles in a post-vaccination setting. The implementation of these local networks, combined with a continued global appreciation of PPRV epidemiology will be key to the long-term sustainability of the PPR eradication campaign.

Other epidemiologically relevant issues such as the characterization of viral propagation rates, the decay rate of maternal antibodies, and age-specific infection and case fatality rates, are also early priorities for avoiding PPR re-emergence in ‘free’ areas.

### 3.5. Infection of Wildlife and Other Species

PPR is a disease of domestic livestock that is a threat to conservation of endangered wildlife small ruminant species. To date, there is no evidence that wildlife can maintain the disease in a way significant to the global epidemiological control of PPR. The possible areas for research at the livestock/wildlife interface have already been extensively reviewed [12]. But is missing in this review, the research on wildlife and non-conventional species as well as the role of wildlife and wildlife/livestock interface in the epidemiology of PPR in different regions in order to address knowledge gaps. Epidemiology of the disease could focus on the potential role of wildlife and other animal species (including non-typical species in surveillance) to inform sheep and goat vaccination plans (PPRV transmission studies using virulent PPRV strains in sheep and goats and also non-typical species), There are now convincing reports demonstrating the ability of PPRV to cross the species barrier. Indeed, PPRV can infect animal species other than small ruminants, with dromedaries, wild goats, pigs and cattle reportedly being identified with PPRV [12,13,14,15,16]. It is currently unclear whether these infections are relevant from an epidemiological and eradication perspective; however, it is essential to fully understand the role of wildlife in the spread and potential maintenance of PPRV in the environment in order to be able to initiate successful control strategies. Along these lines, further support is also needed to help local authorities understand and differentially diagnose PPR from a variety of diseases that cause similar respiratory problems and mortality in small ruminants.

The importance of this research was highlighted by a recent outbreak in Mongolia. In December 2016 the disease was diagnosed in several wildlife populations in the East of the country, including saiga antelope (Saiga tatarica mongolica), ibex (Capra sibirica) and goitered gazelle (Gazella subguttorosa), with more than 80% mortality of the 10,000, highly endangered saiga population of the country [12,16]. Studies in camel populations, kept together with small ruminants, in the Isiolo, Mandera, Marsabit, and Wajir counties of Kenya, established that camels in the study area suffer with PPR manifesting clinical signs that are mainly characterized by inappetence, loss of body condition, and general weakness, leading to diarrhea, conjunctivitis, and ocular nasal discharges preceding death. These clinical signs are similar to those observed in small ruminants with slight variations of manifestations such as keratoconjunctivitis as well as edema of the ventral surface of the abdomen. This shows that camels could be involved in the epidemiology of PPR in the region and that PPRV could be involved in epidemics of disease in camels [2,17,18,19]. Moving forward, an important first step is to ensure that the currently available tests for sero-diagnosis of PPRV are validated in serum samples from these animal species, e.g., camels, saiga and ibex. With specific reference to the PPRV eradication campaign, and as mentioned above, the significance of these infections as a whole should be carefully evaluated, as they may not significantly affect the ultimate success of the programme.

### 3.6. Vaccine Delivery and Novel Vaccines

The current PPR control and eradication strategy includes administration of a live-attenuated vaccine, which following one inoculation provides lifelong immunity to all PPRV lineages as well as detectable antibodies. There is an ongoing debate about the recruitment rate of newly susceptible sheep and goats into small ruminant populations following vaccination, and how this may require more frequent vaccinations than the annual ones that proved so successful during rinderpest eradication. In addition, currently, a lot of vaccination (carried out for political reason in the pastoral areas close to elections) do not reach sufficient levels of immunity to interrupt transmission. The most challenging practical issues are related to the delivery of vaccines to small ruminants at sufficient intensity to generate herd immunity levels that are able to prevent the re-introduction of disease. This could be investigated through estimating the R_0_ value for the PPRV through computer modelling; however, empirical data would be persuasive. Within the pastoral area, there is often a lack of harmonisation of vaccination campaigns at the epizone level. This is also due to the lack of simultaneous funding of PPR activities in neighboring countries. Of note, it was difficult enough to generate these levels of herd immunity for rinderpest, and we must acknowledge the fact that small ruminant numbers are far greater than those of cattle and buffaloes are. Indeed, the virus could be cycling persistently at a low frequency among extensive rural small ruminant populations, much as rinderpest did in eastern African pastoral cattle. There is therefore a need for countries to define precise eradication strategies and implement them in a dynamic manner. The concept of using repeat pulses of extremely efficient vaccination, separated by short time intervals, in defined areas was developed for measles control and was adapted successfully for eliminating rinderpest in Tanzania in 1997 [20]. This technique might be explored further by epidemiological modelling with the hope that it can break virus transmission in endemic areas, when supported by good clinical disease surveillance. Recent mass pulsed vaccination in southern Ethiopia demonstrated the feasibility of rapid area-wide clearance of PPRV infection [21]. A study in Karamoja (Uganda) suggested that integrating the supply chain of PPR vaccine with other veterinary or health commodities could reduce costs, as well as increase uptake [22]. Separately, maintenance of an effective cold chain for this vaccine has also proven difficult in subtropical countries. Implementation of a thermotolerant live-attenuated conventional or recombinant vaccine may represent the best way to avoid cold chain-associated problems in these areas.

Moving forward, we must also admit that although the standard PPR vaccines are cheap and effective, there is a need, especially in developing countries, for vaccines that support broader sustainability in small ruminant production. To this end, there have been significant advances in designing multivalent small ruminant virus vaccines that can also contribute to PPR eradication. Additional field application of these multivalent vaccines to control common small ruminant pathogens, including PPR, would indeed help to enhance poverty alleviation. Considering the enormous benefits of multivalent vaccines, several vaccines are available or currently being developed, e.g., sheep/goat pox and PPR [23].

Whilst risk-based vaccination of sheep and goats in endemic countries might be a pragmatic approach to control PPR in the first phase of disease eradication, the development of a marker vaccine with a robust companion test may help with serological surveillance in the future. Other possibilities include the development of a vaccine capable of differentiating infected from vaccinated animals (DIVA). Emphasis in this area should also be placed on deployment models for these vaccines, as well as their applicability in various hosts (small ruminant and wildlife), to ensure they are utilised to their maximum potential.

### 3.7. Applying Movement Controls and Providing Better Channels for Communication

The control of animal movement, including the imposition of quarantine and other sanitary measures, are integral to most infectious disease control and eradication programmes. However, strict movement control can be counter-productive because it can actually stimulate unnecessary or illegal movement of animals in order to bypass quarantines and restriction orders. To this end, movement controls must be tempered by the experience of local animal health teams who are better equipped to judge the behaviour of local owners when faced with such restrictions.

Whether vaccination is free or not, has been a major concern in several epizones with pastoralists moving in search of free vaccination. If not informed, the vaccination teams will be forced to come back in to the same areas several times to make sure that animals are immunized. Indeed without comprehensive cooperation from stakeholders, any animal control and or disease eradication campaign could ultimately be more costly and take longer to complete or even fail entirely. The level of acceptance of vaccination could be a problem in endemic areas or in areas where the disease is not yet present. In Uganda, a participatory epidemiological assessment included risk mapping with livestock owners, community animal health workers and veterinarians. This indicated that there were two critical foci of virus transmission on the Uganda-Kenya border [24]. Better communication is also essential for developing and influencing people’s attitudes, behaviour and decisions concerning the control of PPR. It is also essential to learn and share knowledge about the attitudes of livestock owners and other key stakeholders. Social science methodologies could be used across the whole eradication campaign to maintain the link between local stakeholders and the eradication team at the regional/national level or to explore potential options such as making herders contribute or not to the vaccination. The absence of communication at this level is another challenge to be addressed.

### 3.8. Applying Serological Monitoring in the Field and the Demonstration of Absence of Disease

According to the objective of the surveillance, it would be useful to use serological monitoring to investigate the levels of recruitment and to develop a proposed methodology for risk-based surveillance, which could be translated into useful actions such as targeted re-vaccination. There are wider calls for serological monitoring to be used to assess the success rates of vaccination campaigns or to invigilate the effectiveness of individual vaccination teams. However, in reality this may only be useful if immediate revaccination can be carried out, which is not often possible. Whilst it is important to identify technical or administrative errors or indeed administrator negligence and address them as the implementation of penalties is very difficult in several countries. Ultimately, the case for detailed serological monitoring may have to be made on a case-by-case basis, depending on a cost-benefit analysis and whether the resultant data can realistically contribute to improve long-term planning. In at risk area that have never declare the disease or when country has stop the vaccination for few years, serology could be carried out to demonstrate the absence of PPRV. In the absence of DIVA and the current situation in several countries, the challenge is the interpretation of positive samples. Careful surveillance strategy taking in to account the vaccination history should be considered.

### 3.9. Accurately Assessing Socio-Economic Impact

Although there are many parameters available to facilitate the evaluation of the socio-economic impact of a disease, there are several drawbacks to applying these, e.g., they can be subject specific or handle only one major factor at a time, therefore lacking the ability to estimate the cumulative impact of a disease on the economy. Nevertheless, these economy-wide considerations are crucial in implementing and funding control and eradication strategies for emerging diseases. For PPR a well-planned cost-benefit analysis of PPR, comparing policies and responses that include both the direct and indirect impacts associated with PPR are needed to better understand the impact of PPR in all settings [25]. That being said, the annual global impact of PPR has been estimated at between USD 1.4 billion and USD 2.1 billion; with cost-benefit studies also carried out in different countries; identifying losses which clearly justify both national and global PPR eradication programmes being pursued [4].

### 3.10. Funding and Political Will 

Many of the countries where PPR is now endemic simply cannot finance an efficient, effective and sustained control/eradication programme. Indeed, they will need significant support for operational costs, training and meetings in order to properly implement PPR GEP. Even if livestock owners themselves contribute more towards the costs of vaccination, there will still be a requirement at the regional and global level for international funding to provide technical and coordination costs, as well as member state support [4,26]. In this context, it will be necessary to explore public-private partnerships, such as those that have proved so effective for polio, measles and malaria control. The PPR GEP budget for 2017–2021, was estimated at USD 996 million. The majority of the funding for PPR GEP relies on resources at a country level, in particular national budgets [4]. The advocacy for fundraising should be supported by socio-economic studies. A recent study in Kenya and Uganda suggested that trade routes, refugee camps and areas of animal crowding during droughts increase PPR incidence as well as the socio-economic impact. These should be targeted for interventions, monitoring and surveillance as part of PPR control programmes [27]. However, global funding is needed for some catalytic components of the programme, including activities that support the effective mobilization of additional national and regional resources. On 6–7 September 2018, FAO and OIE, in collaboration with the European Union and the African Union Commission, organized the global conference “Partnering and Investing for a PPR-free World” in Brussels. The Conference represented a historic milestone for the collective efforts to address PPR. Prior to the conference, a mapping of PPR funding shown that of the approximately USD 1 billion needed for the first PPR GEP phase (2017–2021), USD 656 million have been secured through national budgets and resource partners. The critical financial gap was therefore estimated at USD 340 million [28]. That said, it is worth highlighting that more than 70% of the PPR GEP budget is allocated for the vaccination of 1.5 billion sheep and goats (this includes costs for vaccine procurement, logistics and post vaccination evaluation). It is unfortunate to note that the funding already secured, only small portion of these resources is dedicated to PPR control and eradication. Research is also important to identify indicators for monitoring the eradication programme.

## 4. Conclusions

The ever-advancing spread of PPR has made clear the negative economic impact of the disease, and consequently the benefits of its eradication, protecting the livelihoods of the world’s poorest populations, in line with the United Nations’ Sustainable Development Goals (SDGs). The need for coordinated global action is now becoming stronger than ever. Given the similarities with the rinderpest control programme, the PPR GCES shows early promise and should yield early successes. Specific lessons learnt from PPR GCES in its first 5 years include: (i) PPR monitoring and assessment tools should learn from rinderpest eradication rather than Food and Mouth Disease (FMD); (ii) vaccination should be applied after careful identification of at-risk areas through participatory epidemiology; (iii) vaccination should be followed by an assessment of protection rates so that follow-up vaccination can be planned; (iv) regional/epizone campaigns are needed based on lessons from rinderpest eradication and; (v) more advocacy is needed for fundraising and awareness at grassroots level. PPR is a killer disease, threatening more than 68% of the world’s small ruminant population, and any control programme should quickly gain the approval, agreement and participation of livestock owners in endemic areas. With only one serotype in circulation and a vaccine that provides lifelong protection, an international campaign to eradicate this disease is certain to succeed. Videos, flipchart and other PPR related information can be found at PPR web page [29].

## Figures and Tables

**Figure 1 viruses-12-00313-f001:**
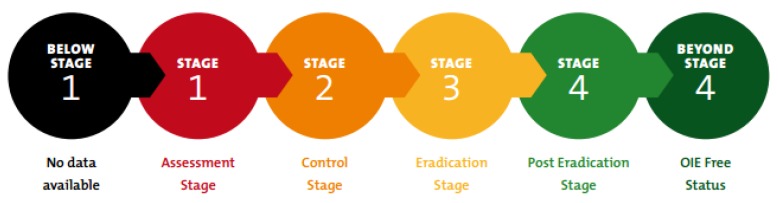
The four stages of the PPR GCES [1].

**Table 1 viruses-12-00313-t001:** Factors important in mounting systematic control programme for PPR (personal communication—findings from a PPR workshop in 2012, Horn of Africa).

Factors	Comments
Small ruminant renew rate understood?	More studies needed
Single serotype?	Facilitates control
Mild form?	Not well known
Carrier state?	Facilitates control
Wildlife reservoir?	Facilitates control
Effective vaccines?	Nigeria 75/1 and Sungri vaccine
Vaccines induce long-lived immunity?	Facilitates control
Vaccines are safe?	Facilitates control
Vaccines are affordable and accessible?	Could be better produced at a large scale
Thermostable vaccines?	Technology available
Quality-assurance systems in place?	AU-PANVAC, as independent vaccines quality control is present in Africa, otherwise no, such quality control exist in Asia
Marked vaccines/DIVA system?	Desirable but not initially important in control programme
Vaccine production SOPs readily available?	Could be developed relatively easily
Clear epidemiological understanding?	Some deficiencies, research required: need not delay initiation of systematic control
Vaccine presentation suitable/packaging/	Small dose vials required
Robust, validated laboratory diagnostic tools for agent detection, and serology to support rapid diagnosis, surveillance and seromonitoring of vaccination?	Further development required
Pen-side rapid test available?	To be validated; not affordable to poor farmers
Laboratory networks to support technology transfer for diagnosis and surveillance?	Existent in only a few regions
World reference laboratories established and supported?	Three PPR WRL exist: CIRAD (France), CAHEC (China), and The Pirbright Institute (UK)
Vaccine delivery optimized?	Use of CAHWs, and animal marking a challenge

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
