# Peer review of "Eradicating the Scourge of Peste Des Petits Ruminants from the World"

_viruses, 2020, doi:10.3390/v12030313_

Round 1

Reviewer 1 Report

It is a well-written review regarding an international program to worldwide eradicate the Peste des Petits Ruminants (PPR), which is a highly contagious viral disease of both domestic and wild ruminants. Some small concerns for the authors to consider during the revision:

More discussion about the impact of PPR in the animal industry in the Introduction is needed. No sufficient information provided per the Applying serological monitoring in the field in Section, and so for The limitation of description about effective vaccination in wild animals Line 18: typos for sentence punctuation after “when” Line 19: add “of” after understanding Line 23: The last three keywords are not topic/scientific specific Line 42: Split the Table title and table body, and references for table 1? Line 105: typo for sentence punctuation after “including” Line 107: add “the” before “disease” Line 141-142: this sentence is not clear, need to clarify.

Author Response

Your concerns have been addressed in the attached version

Reviewer 2 Report

Manuscript focuses on the FAO-OIE joint programme for global eradication of PPR. It summarise the strategy, the steps of implementation and the international coordination.  Furthermore main  scientific, technical and policy challenges are described, that should be addressed to achieve the control of the disease.

The paper is interesting, clear and well written and I recommend its publication. I just have minor suggestions concerning table 1:

the title of the second column is not immediately clear, it may be modified in order to make the meaning more clear. line 16, some more details may be provided in the comment to factor: “Robust, validated laboratory diagnostic tools for agent detection, and serology to support rapid diagnosis, surveillance and seromonitoring of vaccination”   line 4: it is probably “carrier state” the whole table should be edited to be homogeneous to the rest of the document, the font in particular.

Author Response

Your concerns as well as those from other reviewers have been addressed in the attached version

Reviewer 3 Report

Review Viruses : Eradicating the scourge of Peste des Petits Ruminants from the World

General comment

This review manuscript presents the global eradication campaign of PPR, going through its justification, the international mobilization around the topic, the methodology to achieve this objective and provide a list of the major challenges ahead.

This manuscript is well written and provides a good review on the current status of the eradication campaign. The two main issues are that: 1/ its current organization is not optimal with very short sections (e.g. Introduction) and sections that should be merged or re-organised; 2/ it lacks references, some very recent that would strengthen the manuscript. A final comment is that in the challenge section, I have not seen much about the need for social sciences in particular participatory approaches to prepare the ground for vaccination campaign. The level of acceptance of vaccination could be a problem in endemic areas or in areas where the disease is not yet present. Social sciences’ methodology could be used in the whole eradication campaign to maintain the link between local stakeholders and the eradication team at the regional/national level or explore potential options such as making herders contribute or not to the vaccination. Maybe this point could be added in the challenge.

Regarding the first issue, I would suggest to regroup/re-organise the first 2 paragraphs and parts of the 3 paragraphs into an introduction: 1/presenting disease, geographical extent, impact; 2/ Need for eradication, validated by international community & organisations; 3/ Scientific justification. Then I would combine part of paragraph 3 and the whole of 4 into a paragraph presenting the structure  & functioning of the organisation and the Stepwise process, associating as well the small paragraph on Global Vaccination. So up to the section on “Key challenges”, I suggest to have Introduction / Stepwise process & implementation (with possible sub-sections). If possible, an additional figure presenting the different bodies (GCES – GREN – GEP and other actors) and how they function and interact to achieve eradication.

Regarding the references, I have provided a few that could be used in the manuscript.

Specific comments

Abstract

- Need to describe the geographical extent of the disease, briefly in the first sentences

Introduction

- Introduction is very short

- See suggestion in general comments

The scientific justification for eradication

- In order to justify eradication, you can also refer to:

R. Thomson, G. T. Fosgate and M. L. Penrith: Eradication of Transboundary Animal Diseases: Can the Rinderpest Success Story be Repeated? Transbound Emerg Dis (2015)

- Table 1 should more explicit, including its legend. Is the factors’ list an exhaustive list of ref. 1 & 2? Then some factors such like “small ruminant renew rate” à how come it “does not apply” (column 2), as the answer would always be “yes”? Then lines should be organized by thematics (knowledge about epidemiology, surveillance, control)

- L.37: “Importantly …”: this needs to be rephrased: define better “stakeholder partnership at all levels”

The FAO/OIE joint PPR Golbal Control and Eradication Strategy (PPR-GCES)

- Maybe mention the eco-regional approach that is promoted through the eradication programme and is an innovative approach worse mentioning.

Initial implementation of the PPR GCES

- See suggestions in General comments

International coordination of the PPR-GCES

- RAS

The global vaccination strategy for PPR eradication

- This (small) section could also be merged with the 2 previosu ones (international coordination for eradication)

- Maybe provide a breakdown of the number of countries in stage 1, 2, 3, 4 & 5, per continent if possible?

- See for ex. Recent publication on status in southern Africa in a special issue on PPR:

Britton, A. Caron and B. Bedane: Progress to Control and Eradication of Peste des Petits Ruminants in the Southern African Development Community Region. Front Vet Sci, 6, 343 (2019)

Key challenges for the eradication campaign programme

Also mention that the renewal rate of small ruminants (>30% per year) could be a problem to maintain a good immunity at population level This second challenge about research needs is mentioning a lot of the questions/points raised in the other section of this paragraph. This should be avoided and maybe it should concentrate mainly on how research can be embedded in the eradication programme and the different part of the programme it can help (epidemiology, implementation etc.) with a few examples not already presented in the document.

L176-181: provide refs. On the molecular characterisaiton of lineage IV; for ex.:

Mantip, D. Shamaki and S. Farougou: Peste des petits ruminants in Africa: Meta-analysis of the virus isolation in molecular epidemiology studies. Onderstepoort J Vet Res, 86(1), e1-e15 (2019) doi:10.4102/ojvr.v86i1.1677

5. In this section maybe add wording of “non conventional species” for cattle, pig, camels  etc. And add also the ref  Fine et al. above and this one:

Pruvot, A. E. Fine, C. Hollinger, S. Strindberg, B. Damdinjav, B. Buuveibaatar, B. Chimeddorj, G. Bayandonoi, B. Khishgee, B. Sandag, J. Narmandakh, T. Jargalsaikhan, B. Bataa, D. McAloose, M. Shatar, G. Basan, M. Mahapatra, M. Selvaraj, S. Parida, F. Njeumi, R. Kock and E. Shiilegdamba: Outbreak of Peste des Petits Ruminants among Critically Endangered Mongolian Saiga and Other Wild Ungulates, Mongolia, 2016-2017. Emerg Infect Dis, 26(1), 51-62 (2020) doi:10.3201/eid2601.181998

L210-211: “and other African wildlife species”

No mention about the need for international and (eco-)regional coordination to achieve eradication?     

Conclusions

- L303: “poorest populationS”

- L309: “certain to succeed”… few things are certain… and a global eradication campaign may have a high probability of success but could never be certain 10 years before its deadline.

References that could be used in the manuscript:

A. E. Fine, M. Pruvot, C. T. O. Benfield, A. Caron, G. Cattoli, P. Chardonnet, M. Dioli, Y. Dulu, M. Gilbert, R. Kock, J. Lubroth, J. C. Mariner, S. Ostrowski, S. Parida, S. Fereidouni, E. Shiilegdamba, J. Sleeman, C. Schulz, J.-J. Soula, Y. Van der Stede, B. G. Tekola, C. Walzer, S. Zuther, F. Njeumi and Meeting Participants: Eradication of peste des petits ruminants virus and the wildlife-livestock interface. Frontiers in Veterinary Science (2020) M. Kamel and A. El-Sayed: Toward peste des petits virus (PPRV) eradication: Diagnostic approaches, novel vaccines, and control strategies. Virus Res, 274, 197774 (2019) A. R. Cameron: Strategies for the Global Eradication of Peste des Petits Ruminants: An Argument for the Use of Guerrilla Rather Than Trench Warfare. Front Vet Sci, 6, 331 (2019) W. Taylor: The global eradication of peste des petits ruminants (PPR) within 15 years-is this a pipe dream? Trop Anim Health Prod (2016) B. A. Jones, K. M. Rich, J. C. Mariner, J. Anderson, M. Jeggo, S. Thevasagayam, Y. Cai, A. R. Peters and P. Roeder: The Economic Impact of Eradicating Peste des Petits Ruminants: A Benefit-Cost Analysis. PLoS One, 11(2), e0149982 (2016)

Author Response

Your concerns as well s those from other reviewers have been addressed in the attached version.

Reviewer 4 Report

General Comments

The paper has some great moments, but needs work. The topic merits a truly moving presentation. The program is described as it was conceived internally at FAO-OIE, but does not describe any of the learning that has occurred over the last 5 years of implementation. The paper is an excellent opportunity to present the program as a dynamic activity that is evolving in light of information and learning that has been achieved to date. Dynamism was what made rinderpest eradication successful. Every 3 to 5 years the program completely re-invented itself.

What about the ecozones approach? It is not mentioned anywhere. That is a key lesson that will reshape your program.

The overall strategy presents elements of progressive control that does not directly contribute to eradication and may actually compromise the focused activities needed to eradicate PPR. Currently, a lot of the ‘progressive control’ vaccination taking place under stage 2 is at a level insufficient to interrupt transmission. Vaccine is being distributed to administrative units (Districts, etc) on the basis of equity in a mass vaccination approach rather than a refined eradication strategy. The level of herd immunity registered in the population can’t be accounted for by the quantities of vaccine issued and the is actually a reflection of the value of R nought for the wild virus. This is the worst-case scenario. Suppressed disease that is difficult to detect.

The message to countries about the need to define precise eradication strategies and implement them in a dynamic manner is does not come through. The description of the stages and the use of PMAT is bureaucratic at best. How has staging and PMAT worked in the first five years? Has it been well applied and resulted in insights and strong strategies? What are the new directions to be taken.

Epidemiological assessment should be described with greater clarity earlier in the paper. The lines 165-170 do encapsulate this message, but it is fairly well buried. I would suggest using this description in the definition of stage 1 so it is very clear what is needed to complete stage 1. Many countries claim to be in stage 2 or 3, but they don’t have an epidemiological assessment that identifies the sources of infection and critical control points where virus circulation can be interrupted. This is actually feedback that has come through PPR GREN. Genomics should be highlighted as the tool that can show us the linkages between foci of virus circulation, especially which populations are acting as the incubators and sustaining infection in their regions

Section 5: Important message that does not come through: PPR is a disease of domestic livestock that is a threat to conservation of endangered wildlife. To date, there is no evidence that wildlife can maintain the disease.

Specific comments:

Line 12: small ruminants should replace (sheep and goats)

Line 16: capitalize the work organization

Line 42: Table 1 looks like it is the result of a participatory discussion directly transcribed into the paper. I would suggest some editing for clarity and style with an explanation of how the table was created and/or what meeting group created it.

Line 47: FAO and OIE did RP eradication alone? Unacceptable. What about IBAR, IAEA, national governments, the NGOs that did the job in Sudan, research and university partners? I would suggest crediting the international animal health community for the eradication rather than two organizations.

Line 127: Research ‘requirements’ I think all are agreed that these are not ‘requirements’ in the sense that they must be had to finish eradicatio. They are instead ‘nice to haves.’ I would suggest ‘Research to facilitate…’

Line 130: ‘reservoirs’ would better by ‘small ruminant populations acting as reservoirs’

Line 134: Again, nice to haves, but not required: novel vaccines (marked, thermo-tolerant and recombinant, e.g. vaccines able to distinguish infected from naturally infected [DIVA] animals)

Line 134: thermotolerant and thermostable vaccines are not necessarily novel vaccines. The existing vaccine can be and have been packaged in thermostable presentations for example.

Line 170-173: ‘Along these lines evidence of a fully functional disease reporting system (in order to properly quantify the effectiveness of vaccination) or a prolonged period of field work in one or more infected countries may help to generate some of the basic longitudinal data required for computer simulations.’ We all know that many of the high-risk areas for PPR actually have the weakest disease reporting. We know that through expert opinion and experience in the areas as professionals and working with local knowledge. Thus, disease reporting data is unreliable for quantitative analysis. To use it as data, you would need to validate the system. The process of validation would provide more accurate data than the reporting data. I would suggest you focus on targeted short-term studies and appraisals rather than try to rely on notoriously biased and insensitive reporting data.

Line 175: Hmm. Last time I spoke to the authors, this was not a progressive control program.

Line 239: The statement is not accurate. The vaccines are sufficient for the completion of eradication as was the case for rinderpest. Marker vaccines, etc. can contribute. Further, the program should not be undertaking mass vaccination without defined epidemiological goals as would be the case in progressive control.

Line 253: Serological monitoring alone is insufficient. The absence of disease must be demonstrated as well. Currently, a lot of the ‘progressive control’ vaccination taking place under stage 2 is at a level insufficient to interrupt transmission. The level of herd immunity registered in the population can’t be accounted for by the quantities of vaccine issued and the is actually a reflection of the value of R nought for the wild virus.  

Conclusion:

What has PPR GCES learned in the first 5 years? It is a lot and that is the message that needs to come out.

Author Response

Your concerns as well s those from other reviewers have been addressed in the attached version

Round 2

Reviewer 3 Report

Review Viruses : Eradicating the scourge of Peste des Petits Ruminants from the World – Second review

General comment

This review manuscript presents the global eradication campaign of PPR, going through its justification, the international mobilization around the topic, the methodology to achieve this objective and provide a list of the major challenges ahead.

This manuscript is well written and provides a good review on the current status of the eradication campaign. Compared to my previous, the manuscript has gain in clarity and structure. There is still a lack of references in some sections and minor suggestions that have not been addressed in the new version and not discussed in any “response to reviewer”.

Abstract

- L14-16: I would make a new sentence from “In April 2015…” and mention that those international organizations have designed a strategy to eradicate the disease. So far in this abstract, you don’t talk about the strategy that you describe in the main text.

- L18: and elsewhere in the manuscript: “poorest population” in its singular form tends to homogenize human populations across the world. I would preferentially use the plural form “poorest populationS”

Introduction

- L31-32: remove the sentence “now more than around 70 countries in Africa, Asia and the Middle East have reported PPR infections” as you now give more exact figures in the next sentence.

- L32: change sentence as not to start the sentence with “198”

- L34-36: give a FAO reference (FAOSTAT?) for the figures about SR populations.

- L36-39: need a reference for the Turkana study

- L39: “Mongolia reported itS first ever PPR outbreak”

- L39-42: Sentence needs rephrasing as it is not grammatically sound as it stands

- L43: I did not find the 7.27 million figure in the reference number 2. Are you sure it comes from there? If not, can you cite your source?

- Table 1: following my request you have given the source of the Table. My earlier comments still stand: the table should more explicit, including its legend. The title mentions factors important for a control programme against PPRà how come some factors do not apply? Maybe the second and third column should be merged as this second column confuses me.

FAO/OIE joint PPR GCES

- Ref to my previous comment: “Maybe provide a breakdown of the number of countries in each stage 1, 2, 3, 4 & 5, per continent if possible” You did not reply if this data is available (and should be presented to showcase the global stage of eradication process as of 2020) or if this data is not available. I think this is important data for the manuscript to indicate the amount of “work” still ahead.

- L100 : If there is a PMAT manual that exists, it should be referred to here. I think this is important for the reader.

- L108-109: is there a report mentioning the creation of the GREN in 2018? This could be referred to.

- In this section, as commented earlier, if possible, an additional figure presenting the different bodies (GCES – GREN – GEP and now Friends if PPR GEP other actors such as national and regional programmes) and how they function and interact to achieve eradication would be welcome. I don’t think I have ever seen that in OIE/FAO reports and it would be a good output of this article.

- L144-145: “identification tool”: any reference about this information?

- L145-148: cite the source of the information from this sentence if not ref 5.

Key challenges

- Challenge SR production: I think you should emphasize that the short cycle nature of production with an estimated annual turnover rate of 30% is an important factor/challenge to consider regarding the vaccination programme.

- Challenge Research: as mentioned previously, this second challenge about research needs is mentioning a lot of the questions/points raised in the other section of this paragraph. This should be avoided and maybe it should concentrate mainly on how research can be embedded in the eradication programme and the different part of the programme it can help (epidemiology, implementation etc.). For example, you mention “filter paper sampling

for serology; rapid field tests”: this is too much detail for this paragraph and should be addressed in the relevant Challenge on diagnostics. In addition, you should at least mention research on wildlife and non-conventional species” as there is a challenge about this and research will play an important part in it. Research is also maybe important to monitor the eradication programme. For Rinderpest, we are lucky today to benefit from people involved in the eradication programme, but I feel that the articles referring to the Rinderpest eradication programme, successes and failure are lacking. Maybe w ecould avoid that for PPR.

- Challenge Laboratory: references are all packed in one sentence when they should be put in the right part of the text. And more references on diangostics are needed here (for ex. On genomics analysis: there are a few recent papers that have brought interesting insights on this aspect).

- Challenge epidemiology: there are no references in this paragraph and it refers to many studies? They should be added.

- L137: “low RISK”

- L223-241: I am not sure this section should be in this paragraph. Maybe you can move it up to the section 2.1 when presenting the step-wise framework.

- L242-250: need many references here as you refer to many studies.

- L262-264; need references

- Challenge wildlife L285-287: already said at the beginning of the paragraph

- Challenge Vaccine L296: “Being sometime in the pastoral area,…” : rephrase

- As mentioned above, the fact that each year around 30% of your SR population is renewed with naïve individuals is not mentioned as a challenge for vaccination?

- Challenge Movement control: the new title should be rephrase as it does not mean much in its current form. The whole paragraph should be reworked.

- L336-338: I don’t understand boths sentences.

- L345: I don’t understand the meaning of “communication for development”. Please define or provide a ref.

- Challenge serological monitoring: here you mention the recruitment rate of SM and its impact on vaccination. I would have mentioned this in the vaccination challenge.

- L356: define better “political vaccination” and “wild virus”.

- L357-358: sentence should be re-written grammatically.

- L355-368 : need references

- L414: I reiterate my caution: “certain to succeed”… few things are certain… and a global eradication campaign may have a high probability of success but could never be certain 10 years before its deadline.

- L414-419: those added line should be put elsewhere in the conclusion paragraph and not after the previous conclusive sentence.

References

FAO and OIE: Partnering and investing for a peste des petits ruminants-free world. In: Partnering and investing for a peste des petits ruminants-free world. Ed FAO&OIE. Brussels, Belgium (2018)

Author Response

- L108-109: is there a report mentioning the creation of the GREN in 2018? This could be referred to.

Response: There is only meeting recommendation that is not yet in the PPR web page.

- In this section, as commented earlier, if possible, an additional figure presenting the different bodies (GCES – GREN – GEP and now Friends if PPR GEP other actors such as national and regional programmes) and how they function and interact to achieve eradication would be welcome. I don’t think I have ever seen that in OIE/FAO reports and it would be a good output of this article.

Response: The next paper is under preparation and should include the governance of the programme.

- L144-145: “identification tool”: any reference about this information?

Response: This was suggested during one of the road map meeting. But need to be adapted by all the 9 regions.

for serology; rapid field tests”: this is too much detail for this paragraph and should be addressed in the relevant Challenge on diagnostics. In addition, you should at least mention research on wildlife and non-conventional species” as there is a challenge about this and research will play an important part in it. Research is also maybe important to monitor the eradication programme. For Rinderpest, we are lucky today to benefit from people involved in the eradication programme, but I feel that the articles referring to the Rinderpest eradication programme, successes and failure are lacking. Maybe w ecould avoid that for PPR.

Response: for serology; rapid field tests”: this is too much detail for this paragraph: was requested by one of the reviewer.

- L223-241: I am not sure this section should be in this paragraph. Maybe you can move it up to the section 2.1 when presenting the step-wise framework.

Response: This was requested by one reviewer as the role of epidemiology on the step-wise framework.
